# Time-Dependent Mechanical Properties in Polyetherimide 3D-Printed Parts Are Dictated by Isotropic Performance Being Accurately Predicted by the Generalized Time Hardening Model

**DOI:** 10.3390/polym12030678

**Published:** 2020-03-19

**Authors:** A. G. Salazar-Martín, A. A. García-Granada, G. Reyes, G. Gomez-Gras, J. M. Puigoriol-Forcada

**Affiliations:** Grup d’Enginyeria en Producte Industrial, (GEPI), Institut Químic de Sarrià, Universitat Ramon Llull, Via Augusta 390, 08017 Barcelona, Spain; antoniosalazarm@iqs.url.edu (A.G.S.-M.); guillermo.reyes@iqs.url.edu (G.R.); giovanni.gomez@iqs.url.edu (G.G.-G.); josep.puigoriol@iqs.url.edu (J.M.P.-F.)

**Keywords:** creep, stress relaxation, fused deposition modelling, polyetherimide, process parameters, mathematical characterization, generalized time hardening model

## Abstract

The Fused-Deposition Modelling (FDM) technique has transformed the manufacturing discipline by simplifying operational processes and costs associated with conventional technologies, with polymeric materials being indispensable for the development of this technology. A lack of quantification of viscoelastic/plastic behavior has been noted when addressing FDM parts with Polyetherimide (PEI), which is currently being investigated as a potential material to produce functional end-products for the aerospace and health industry. Primary and secondary creep along with stress relaxation tests have been conducted on FDM PEI specimens by applying stresses from 10 to 40 MPa for 100 to 1000 min. Specimens were 3D printed by varying the part build orientation, namely XY, YZ, and XZ. Creep results were fitted to the Generalized Time Hardening equation (GTH), and then this model was used to predict stress relaxation behavior. FDM PEI parts presented an isotropic creep and stress relaxation performance. The GTH model was proven to have a significant capacity to fit viscoelastic/plastic performances for each single build orientation (r > 0.907, *p* < 0.001), as well as a tight prediction of the stress relaxation behavior (r > 0.998, *p* < 0.001). Averaged-orientation coefficients for GTH were also closely correlated with experimental creep data (r > 0.958, *p* < 0.001) and relaxation results data (r > 0.999, *p* < 0.001). FDM PEI parts showed an isotropic time-dependent behavior, which contrasts with previous publications arguing the significant effect of part build orientation on the mechanical properties of FDM parts. These findings are strengthened by the high correlation obtained between the experimental data and the averaged-coefficient GTH model, which has been proven to be a reliable tool to predict time-dependent performance in FDM parts.

## 1. Introduction

Fused-Deposition Modelling (FDM) is the most extensively used Additive Manufacturing (AM) technology and has recently generated increased interest in various industrial areas, from biomedical to aerospace engineering, by limiting the preliminary cost of driving Computer-Aided Design (CAD) models from the laboratory bench to testable prototypes and by enabling companies to manufacture more complex and functional end-products without the inherent limitations of conventional technologies such as injection molding or embossing [1,2,3,4].

Nonetheless, the essential attribute of this technology includes the capacity to reduce the manufacturing operations required for the fabrication of industrial products [5]. FDM workflow begins by converting CAD models to Standard Triangle Language (STL) format which simplifies the 3D model to a surface geometry, which will limit later the ability of building paths defined in slicer programs to define the numerical code that will control FDM machines. Those machines feed raw materials as a filament into the extrusion head, heating them until they reach their transition temperature. Then, melted materials are extruded through a nozzle, which is moved by two perpendicular, numerically controlled actuators, depositing the material in successive layers to build the desired geometry. Deposited filaments are cooled and solidified at the same time, enabling the union between layers [5].

Since FDM consists of the extrusion of polymeric materials, understanding the mechanical properties of these materials and their interaction with FDM process parameters has become a topic of increasing interest among the scientific community due to industry needs [6]. Research on this topic was conducted in previously published works for different sort of plastics, with Polycarbonate (PC) or Acrylonitrile Butadiene Styrene (ABS) being the most studied because of their wide application in industrial products and the feasibility for them to be acquired in research groups due to their reduced cost [6].

Considerable attention has been given in the literature to the directional orientation in the building process of FDM parts and their properties because it directly impacts the polymer chains’ orientation, which is crucial for understanding the mechanical behavior of potential end-products [6]. Several building parameters are used to define the FDM manufacturing process, namely the part build orientation, which defines the orientation of the part with respect to the planar domain of the nozzle (see Figure 1); the raster angle, which characterizes the orientation of inner filaments within the part with respect to the principal manufacturing axis; or the number of contours, which represents the number of adjacent filaments that follow the outer shape of the printed specimen. Domingo-Espin et al. [7], Croccolo et al. [8] and Galeja et al. [9] studied the tensile properties of PC and ABS parts when varying the layer orientation, raster angle, and number of contours. Ziemian et al. [10,11], and Li et al. [12] described how the air gap and raster angle affect the tensile performance of ABS specimens. Sood et al. [13], Ahn et al. [14], and Montero et al. [15] focused on the compression characterization of ABS parts when modifying the air gap, raster angle, raster width, layer thickness, layer orientation, and printing temperature. Dawoud et al. [16], Mohamed et al. [17], Vega et al. [18], and Lee et al. [19] described how the layer orientation, air gap, and raster angle affect the bending and impact strengths in FDM parts. The torsion behavior of ABS parts when modifying the air gap, raster angle and width, or printing temperature was addressed by Rodríguez et al. [20]. Puigoriol-Forcada et al. [21] and Gomez-Gras et al. [22] studied the cyclic fatigue performance of Polylactic Acid (PLA) and PC specimens when varying the part build orientation.

Meanwhile, time-dependent mechanical properties [23,24,25,26,27] or residual stress [28] have been addressed in fewer studies. The former is indispensable for the validation of end-product parts, since polymer-based materials show a viscous performance that implies a further understanding of time-dependent applications such as clamping pins or mechanical sealing.

Mohamed et al. [23,24] established the relationship between flexural creep behavior of FDM parts and several mesostructures, i.e., the raster angle, layer thickness, air gap, number of contours, and raster width. PC-ABS specimens were tested with a unique load condition for 300 min using a definitive screening design. A quadratic model was used to describe creep behavior varying process parameters, although neither stress nor temperature was considered. The results showed a superior creep resistance performance from specimens in which rasters were arranged in the same direction as the load, and they had a null air gap value with a slice thickness of 0.254 mm.

Kozior and Kundera [25] studied ABS specimens to determine how part build orientation affects relaxation behavior under uniaxial compression. A standard linear solid model was used to fit the experimental data and thus describe the rheological properties. Part build orientation was shown to be a key parameter influencing, in particular, the rheological properties of the FDM parts.

Flexural creep modulus was characterized by Türk et al. [26] in FDM ABS parts. Three-point bending creep tests were carried out when modifying the part print orientation. Significant anisotropic behavior was found in FDM ABS specimens due to layer orientation during fabrication.

Salazar-Martín et al. [27] investigated creep behavior when modifying the number of contours, raster to raster air gap, and part build orientation of FDM PC parts. Different levels of stress held for up to 300 and 1000 min were experimentally addressed. Results showed that these process parameters are significant factors when analyzing the creep behavior of FDM parts, particularly that of part build orientation. Two analytical models were implemented. The fractional Voigt–Maxwell in series was shown to provide better accuracy than the Bailey–Norton equation, despite requiring a longer computational time.

Relevant contributions to creep and stress relaxation behavior in FDM parts are summarized in Table 1. Hence, time-dependent mechanical properties were proven to be a factor to take into account when manufacturing FDM polymer-based parts.

Furthermore, to the knowledge of authors, it is noted that the time-dependent mechanical properties of FDM Polyetherimide (PEI) parts have not yet been addressed in previous works. This material is gaining interest in the aerospace and health industry because of its reasonable mechanical strength and heat resistance and the feasibility of it being manufactured with AM technologies [2].

Thus, this study investigates the primary and secondary creep and the stress relaxation characteristics of FDM PEI parts by varying the part build orientation parameter, which has been proven to be a critical parameter, as discussed above. This could elucidate the iso-/aniso-tropic behavior of PEI in FDM applications and allow us to draw conclusions about their behavior when contrasted with previously studied FDM materials, such as ABS or PC.

Furthermore, since an accurate model to predict the viscoelastic and viscoplastic performance of FDM parts is needed to validate further Finite Element Method (FEM) simulations of FDM geometrical complex parts, the Generalized Time Hardening (GTH) model is evaluated. The model is used to fit the experimental creep data and then used to predict stress relaxation results to validate it.

In addition, due to the lack of data for specific part build orientations, uniaxial tensile tests are addressed in this study prior to long-term endurance tests.

The analytical derivation is presented and discussed in Section 2. Subsequently, the experimental procedure used for the characterization of the creep and stress relaxation test carried out in this work is presented. In Section 4, the experimental results of the investigation are shown and explained. Finally, conclusions are discussed along with their translation to industry and research application.

## 2. Analytical Approach

Nowadays, mathematical and computational models of viscoelastic materials are a topic of hard debate and scrutiny. Many viscoelastic and viscoplastic models have been defined to perform stress and strain analysis on both creep and stress relaxation data [29]. The classical models of Voigt and Maxwell to fractional models or Prony series have been widely used for this purpose [30]. Regarding FDM analysis, Kozior and Kundera [25] used the Standard I model to describe the rheological properties of ABS specimens. A good correlation between experimental stress relaxation data and the Standard I model was achieved. Nevertheless, only one strain level was tested. Modeling of PC FDM parts for different stress levels was conducted by Salazar-Martín et al. [27] using the Bailey–Norton law and the fractional Voigt–Maxwell in series. The former equation presented a good agreement, particularly for low stresses; meanwhile, the latter showed better accuracy, although more parameters were required to accurately predict creep behavior. The nonlinear behavior of FDM parts was reported under moderate stress levels and standard conditions. Other authors used quadratic regression models to predict the creep phenomenon, where process parameters were variables of these models. Neither of these models considered stress, strain, or temperature as function variables [23,24,26].

Because nonlinear viscoelastic/plastic behavior in plastics are expected to be exhibited in prior studies, as in work by Salazar-Martín et al. [27], and because finding a practical approach for predicting creep and stress relaxation behavior in Finite Element Method (FEM) simulations is the aim of this study, the Generalized Time Hardening (GTH) model was chosen [31,32]. This model is capable of explaining the creep behavior of different materials in contrast to time hardening and strain hardening equations more accurately, since GTH is able to model materials’ behavior in which creep compliance is not a constant value in the studied domain, extending its capability from viscoelastic to viscoplastic characterization [31,32]. For instance, in the GTH model, the coefficients are polynomial functions of stress, whereas, in the time hardening equation, these coefficients are constants. Furthermore, this model is implemented in FEM software, such as ANSYS, and is capable of simulating primary and secondary creep. Fractional models were discarded due to fitting complexity and FEM implementation [27].

Furthermore, Garcia-Granada [33] discussed the possibilities of modeling creep. A material model based on previous strain and strain rates was discussed as a strain hardening model and compared to a simplified model where stress and strain were a function of time, named time hardening. This mathematical approach becomes indispensable for FEM simulations when parameters are obtained according to the true strain and strain rates.

The GTH model used in this approach is expressed as [31,32]
(1)$$\left\{\begin{array}{c} \varepsilon_{c}=ft^{r}e^{-C_{6}/T}\\ f=C_{1}\sigma +C_{2}\sigma^{2}+C_{3}\sigma^{3}\\ r=C_{4}+C_{5}\sigma \end{array}\right.$$
where $C_{x}$ are material constants independent of stress, $\varepsilon_{c}$ is the equivalent creep strain, σ is the equivalent stress, *t* is the time, and *T* is the temperature. Values of $C_{x}$ will be affected by process parameters, since they modify the overall mechanical behavior of the part. Coefficients were obtained employing the Levenberg–Marquardt method, which is used to solve nonlinear least-squares problems. The exponential term was neglected because tests were carried out at the same temperature. Thus,
(2)$$\varepsilon_{c}=ft^{r}$$

In order to prove the adequacy of this model, the stress relaxation equation for the GTH model should be deduced. Because strain is constant over the time, it can be expressed as constant strain rate
(3)$$\frac{\partial \varepsilon_{c}}{\partial t}+\frac{\partial \varepsilon_{i}}{\partial t}=0,$$
where $\varepsilon_{i}$ is the initial strain rate. In this case, each summand is
(4)$$\frac{\partial \varepsilon_{i}}{\partial t}=\frac{\partial \sigma }{\partial t}\frac{1}{E}$$
(5)$$\begin{array}{c} \frac{\partial \varepsilon_{c}}{\partial t}=\left(C_{1}\frac{\partial \sigma }{\partial t}+2C_{2}\sigma \frac{\partial \sigma }{\partial t}+3C_{3}\sigma^{2}\frac{\partial \sigma }{\partial t}\right)t^{C_{4}+C_{5}\sigma }+\left(C_{1}\sigma +C_{2}\sigma^{2}+C_{3}\sigma^{3}\right)\;\times \;\\ \;\times \;\left[\left(C_{4}+C_{5}\sigma \right)t^{C_{4}+C_{5}\sigma -1}+t^{C_{4}+C_{5}\sigma }ln\left(t\right)C_{5}\frac{\partial \sigma }{\partial t}\right], \end{array}$$
where E is the Young’s modulus. Taking Equations (3)–(5), the stress rate can be expressed as
(6)$$\frac{\partial \sigma }{\partial t}=\frac{-a\;\left(C_{4}+C_{5}\sigma \right)\;t^{C_{4}+C_{5}\sigma -1}E}{b\;E\;t^{C_{4}+C_{5}\sigma }+a\;ln\left(t\right)\;C_{5}\;E\;t^{C_{4}+C_{5}\sigma }+1},$$
where
(7)$$\left\{\begin{array}{c} a=(C_{1}\sigma +C_{2}\sigma^{2}+C_{3}\sigma^{3})\\ b=(C_{1}+2C_{2}\sigma +3C_{3}\sigma^{2}). \end{array}\right.$$

The Matlab ode23 function was used to operate Equation (Equation 6).

## 3. Experimental Procedure

### 3.1. Specimens and Materials

ASTM D638-02a (2013) was used as the standard to prepare specimens for tensile, creep, and stress relaxation testing [34]. These specimens have a size of 165 × 13 × 7 mm. The material used for the specimens was ULTEM 9085, a PEI designed for rapid prototyping technologies as well as to fulfill aerospace industry requirements [35]. This material is a high-performance thermoplastic with a reasonable strength-to-weight ratio and flame, smoke, and toxicity rating [35]. Stratasys Fortus 400 mc was used to manufacture the samples. Slicer software Insight was used to slice specimens, which were manufactured in groups of up to 40 in the same material and foundation sheet within a temperature-controlled chamber. Parts were manufactured over a polymeric disposable tray fixed on an aluminum plane table without special treatments.

### 3.2. Process Parameters

Mechanical properties are affected by the process parameters and the interactions between them. Therefore, it was a challenging task to define the proper process parameters for the research [6].

Some important mechanical characteristics that are severely affected by process parameters are the bonding behavior, the orientations of the filaments regarding the directions of the applied load, and the raster length or the cross-sectional area ratio between contour filaments and rasters [27]. For instance, the bonding behavior, which is related to the internal cohesion forces among the polymer chains and inter-layer forces, is modified when varying process parameters, thus influencing the global effect of tensile, flexural, or impact strength [27]. Therefore, the study of the parameters with the greatest influence on mechanical behavior is necessary to understand the inner anisotropy of FDM parts arising from the layer orientation during fabrication.

The mechanical properties of FDM parts were proven to be highly modified when modifying the air gap, raster angle, and the number of contours [7,8]. In contrast, raster width and raster thickness have little influence on these properties. Nonetheless, the process parameter that has a large effect on the manufacturing process is the part build orientation [7,8]. Not only the manufacturing time and support material are affected by this parameter, but the mechanical characteristics mentioned above are also severely modified as well as the mechanical properties.

Accordingly, this study establishes the functional relationship between the part build orientation, namely XY, YZ, and ZX (Figure 1), and time-dependent mechanical properties. The isochronous creep test and stress relaxation test were performed to determine the viscoelastic and viscoplastic behaviors of PEI FDM parts.

Sood et al. [36,37] and Ziemian [11] showed that the values stated in Table 2 provide good results in terms of dimensional accuracy, surface finish, and mechanical behavior in any given part build orientation. Therefore, these values have been widely accepted as optimal parameters, and slicer software implements them by default.

### 3.3. Tensile, Creep, and sTress Relaxation Tests

Prior to stress relaxation and creep tests, yield and ultimate stress were identified in the XY direction through tensile tests. No other directions were tested since these data were provided by Stratasys [35]. Five specimens were tested in the XY configuration at 23 °C (room temperature (RT)). An MTS Insight Electromechanical 100 kN machine was used to perform tests. Strain data were recorded from the axial MTS 634.12F-54 extensometer.

For the creep test, an isochronous creep experiment was conducted according to ASTM D2990-17 [38]. First, a stress level of 10 MPa was applied for 100 min. The stress was then removed for a period of 400 min. This procedure was repeated with the same specimen three times more at 20, 30, and 40 MPa. This test was repeated twice for each orientation. Table 3 summarizes the design of the experiments for the creep tests.

In the stress relaxation tests, the same four levels of initial stress were applied. In this case, for each part to build orientation and level of stress, two specimens were tested. Each level of initial stress was applied for 300 min and achieved strain was held constant over time, except for the 20 MPa level, in which the strain was held for 1000 min. This longer test enabled us to prove the validity of the equations presented above for longer periods. Table 4 summarizes the design of the experiments for the stress relaxation tests.

## 4. Results

### 4.1. Elastic and Plastic Response in Tensile Loading

Tensile tests in FDM PEI specimens (Table 5) revealed isotropic material behavior in the elastic domain of ULTEM parts (mean Young’s modulus 2.16 ± 0.09 GPa). XZ exhibited a slight stiffer performance, presenting an elastic modulus 10% higher than the XY direction and 5.6% higher than the YZ direction. On the contrary, significant differences were displayed in the plasticity field, with the YZ yield strength appearing 1.42–1.47-fold higher than the XZ and XY orientations and the YZ ultimate strength being 1.43–1.64-fold higher than XY and XZ, respectively. Moreover, the minimum ultimate strength (XZ: 42 MPa, Table 5) served to limit the scope of the stress analysis in the creep and stress relaxation tests (Table 3 and Table 4).

Minimum differences were found in the achieved infill pattern (Table 3) with an average infill of 87.12%. No correlation was found between the infill and yield strength (P = 0.829).

Although the linear elastic modulus has been used to characterize FDM polymeric parts based on manufacturer’s data and previous publications [35], limited capacity to predict the stress–strain relation within the linear region was found in all printed geometries, specifically when reaching tensile strength (Table 5, Figure 2). For instance, the YZ direction (yield strength: 42 MPa) reported a 1.33-fold higher strain than that predicted with the manufacturer’s data in the initial creep and stress relaxation tests at 40 MPa.

Furthermore, high variability in the initial strain was found when reaching the plasticity domain (XY: SD = 2.24; XZ: SD = 1.52, at 30 MPa).

### 4.2. Creep Response and Model Fitting

The creep test results (Figure 3) mirrored the mechanical behavior presented in the elastic domain for the different part build orientations. Under the viscoelastic regime, the averaged strain achieved for all orientations after 100 min was 2.19$\;\times \;10^{-2}$ ± 6.4% at 30 MPa (Table 6).

Meanwhile, differences between part build orientations for the viscoplastic domain were evident in terms of the averaged creep strain at 30 MPa with 8.01$\;\times \;10^{-2}$ ± 20.7%, because of the instantaneous plastic response reported in the loading step for XY and XZ orientations (Table 6, Figure 3).

Recovery after load removal was not fully achieved as a consequence of permanent deformation induced by viscous flow in loading time, specifically at high levels of stress (≥25 MPa, Table 6).

All creep results were fitted to the GTH model (Figure 3). Individual part build orientation fittings provided a significant capacity to elucidate viscoelastic/plastic behavior for each configuration (XY: r = 0.907, *p* < 0.001; YZ: r = 0.980, *p* < 0.001; XZ: r = 0.927, *p* < 0.001), as depicted in Figure 3.

Despite the inherent anisotropic plastic behavior due to the specimen printed-orientation, averaged coefficients of the GTH model were tightly correlated with all tested configurations at different stress levels (XY: r = 0.962, *p* < 0.001; YZ: r = 0.977, *p* < 0.001; XZ: r = 0.958, *p* < 0.001). Visual inspection of the creep compliance modulus (Figure 4), which was severely affected by the level of stress, confirmed the capacity of the GTH model with averaged coefficients to predict viscoelastic behavior, although the viscoplasticity performance was more limited to the YZ direction. This is because YZ showed an intermediate behavior between the three orientations and, thus, similar coefficients to the mean parameters’ values (Table 7). Furthermore, YZ presented a wider plastic regime to analyze viscoplasticity than the XY and XZ orientations (Table 5).

Numerical prediction of PEI creep behavior withstanding 16 MPa was computed with non-averaged coefficients for the GTH model (Figure 5). PC data provided by Salazar-Martin et. al. [27] gave a mean viscoelastic strain after 100 min of ∼9 mm/m within a range of ∼1 mm/mm, which was compared with the PEI results (mean ∼9.2 mm/m, range ∼0.2 mm/m; Figure 5).

### 4.3. Experimental and Simulated Stress Relaxation Response

Overall, applied stress on tensile specimens decreased significantly by ≥20% within 300 min of load application (Figure 6). An instantaneous relaxation occurred within the first 10 min, decreasing stress values by ≥10%, and this was followed by a smooth molecular rearrangement, driving a limited stress relaxation.

Analytical modelling (Equation (Equation 6)) was able to explain stress relaxation with the coefficients fitted with creep results for each geometry by compellingly correlating (XY: r = 0.998, *p* < 0.001; YZ: r = 0.999, *p* < 0.001; XZ: r = 0.998, *p* < 0.001) both time-dependent behaviors. Surprisingly, this correlation was even improved when the GTH model was computed with mean coefficients (XY: r = 0.999, *p* < 0.001; YZ: r = 0.999, *p* < 0.001; XZ: r = 0.998, *p* < 0.001). Furthermore, long-term assays also proved the validity of the model for longer time periods (Figure 7).

## 5. Discussion

### 5.1. Elastic and Viscous Effects in FDM PEI Parts are Dictated by Isotropic Behavior, not Anisotropic Behavior

Slight differences were observed when contrasting elastic behaviors for different part build orientations. Although XZ showed the highest Young’s modulus (Table 5, Figure 2), this orientation was clearly penalized in the plasticity domain, since, in a uniaxial test, the load is transmitted through the interlayer material (see layer configuration in Figure 1), where the internal cohesion force was proven to be the lowest among the specimens tested with other FDM materials [21,22]. In contrast, the YZ orientation reported the highest yield and ultimate strength (Table 5, Figure 2), because more filaments, in this case, contours, were oriented in the direction in which the specimen was being pulled. This led to a stronger part wall, distributing the residual stress accumulation in wider areas of the specimen. Although XY also exhibited this geometry, the cross-section contours corresponding to the total area ratio were smaller than in the YZ configuration, which led to limited mechanical properties.

Overall, differences in the elastic modulus (2.16 ± 5% GPa, Table 5) and creep strain in the elastic domain (2.19 $\times \;10^{-2}$ ± 6.4%, Table 6) suggest that PEI provides an isotropic behavior in the viscoelastic domain. This finding is strengthened by the high correlation obtained between the experimental data and the averaged-coefficient GTH model for the creep data (XY: r = 0.962, *p* < 0.001; YZ: r = 0.977, *p* < 0.001; XZ: r = 0.958, *p* < 0.001) and the stress relaxation results (XY: r = 0.999, *p* < 0.001; YZ: r = 0.999, *p* < 0.001; XZ: r = 0.998, *p* < 0.001). Moreover, the comparison between PC and PEI (Figure 5) clearly proves the anisotropic performance of other FDM materials [27] versus PEI, reaching PEI specimens in a 5-fold lower range in terms of viscoelastic strain than PC parts at a given load.

### 5.2. GTH Model Efficacy to Fit Creep Results and Predict Stress Relaxation Behavior

The capacity to accurately fit creep data (r≥0.907, *p* < 0.001, Figure 3, Figure 4 and Figure 5) and estimate stress relaxation performance (r≥0.998, *p* < 0.001, Figure 6 and Figure 7) by the Generalized Time Hardening (GTH, Equation (Equation 1)) model provides a precedent for easy time-dependent mechanical characterization for fused-deposition modelling (FDM), which could also be extended to other additive manufacturing techniques (AM).

The better predictive performance of the GTH model compared with other well-known phenomenological viscoelastic models included in FEM packages, such as time hardening or strain hardening, could be predicted because of the creep compliance which is dependent on applied stress [31,32]. The GTH model has the ability to effectively adapt to the increase in compliance modulus (r in Equation (Equation 1), Figure 4), whereas the models mentioned above are unable to exhibit different values for creep compliance in a given material for different stress levels.

### 5.3. Limitations

The linear elasticity hypothesis has been shown to provide a limited capacity to predict the stress–strain relation in FDM PEI specimens. Therefore, a hyperelastic approach is required when addressing the material characterization of FDM parts with demanding loads (≥25 MPa for PEI) to prevent overestimating the material performance (Figure 2).

Because of this stated limitation, the Young’s modulus in Equation (Equation 7) should be carefully computed to avoid computing a biased level of elastic strain (initial loading). Thus, the elastic modulus should be computed for each specimen at the stress level of interest, or a hyperelastic model should be implemented to predict the stiffness increase along with the GTH model (Equations (1) and (6)) to ensure proper estimation of the viscoelastic/plastic performance.

### 5.4. Translational and Industry Implications

FDM PEI parts are being widely studied as functional parts in the aerospace industry, even integrated with composite structures [3,4]. Furthermore, because of its heat resistance characteristic, ULTEM 9085 has been proven to be a reliable material for the manufacturing of medical devices, which needs to withstand high-temperature sterilization cycles [2]. Therefore, elucidating the time-dependent behavior of FDM PEI parts is indispensable for further understanding the capacity of this material to carry out demanding load applications over time, such as in clamping pins or mechanical sealing.

Two main findings were reported: isotropic viscoelasticity in ULTEM parts and the high efficacy of GTH in predicting viscoelastic/plastic behavior. This could serve as a precedent in FDM technology, because of the high anisotropy found in other materials, such as PC and ABS, as well as the capacity to mathematically characterize creep and stress relaxation behavior with a model already integrated in FEM package software, thereby limiting the computational effort and time required to predict the mechanical behavior of complex geometries required for industry applications. Furthermore, to the knowledge of authors, this is the first study with FDM specimens to successfully fit a time-dependent model and validate it with different tests.

Future work will address the matter of testing multiaxial load simulations, complex geometries, and the influence of temperature in creep and stress relaxation tests in FDM parts.

## Figures and Tables

**Figure 1 polymers-12-00678-f001:**
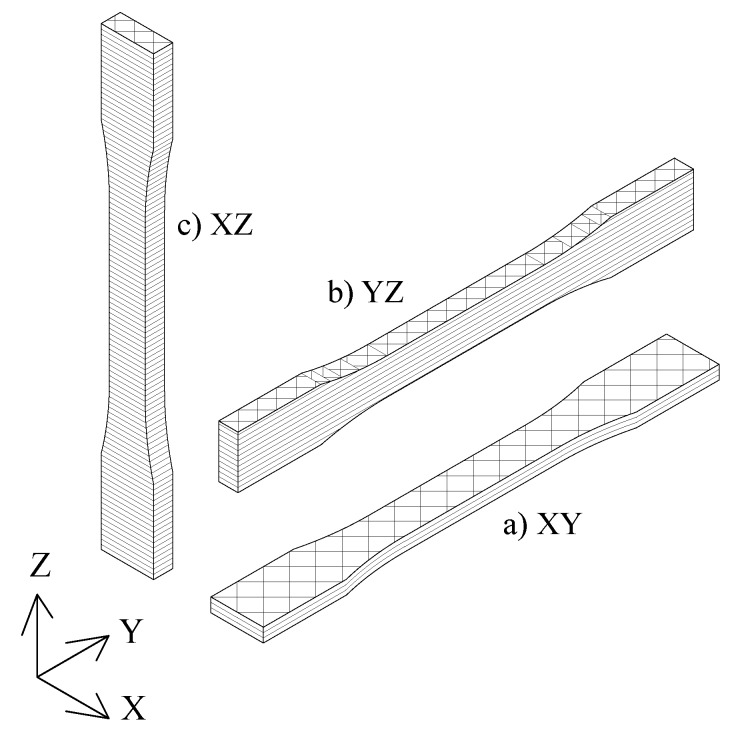
Representative raster and layer configuration for different part build orientations, namely XY, YZ, and XZ. During manufacturing, melted filaments were deposited in the build platform normal to the Z direction.

**Figure 2 polymers-12-00678-f002:**
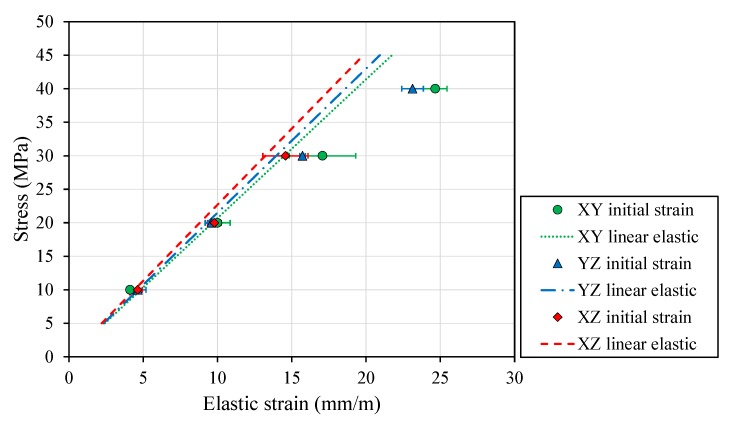
Predicted linear elastic behaviour (dashed lines) based on previous publications (YZ, XZ) [35] and tensile tests (XY). The initial elastic and plastic strain in the creep and relaxation tests (markers) were underestimated by the linear model.

**Figure 3 polymers-12-00678-f003:**
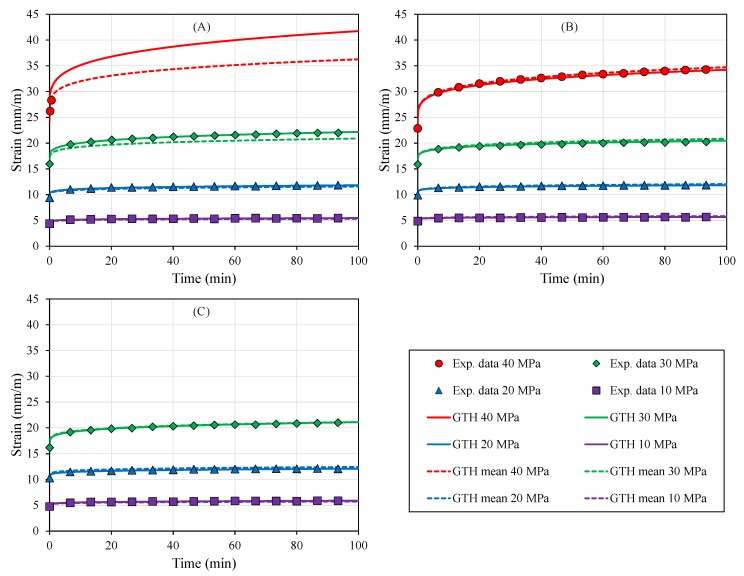
Viscoelastic/plastic curves for different part build orientations at room temperature (RT): (**A**) XY, (**B**) YZ, and (**C**) XZ. The XY and XZ orientations’ specimens broke during the initial loading to 40 MPa. Generalized Time Hardening (GTH) coefficients fitted to each orientation (solid lines) were able to fit creep behavior in all orientations (XY: r = 0.907, *p* < 0.001; YZ: r = 0.980, *p* < 0.001; XZ: r = 0.927, *p* < 0.001), although the averaged GTH coefficients (dashed lines) also provided a reasonable fitting (XY: r = 0.962, *p* < 0.001; YZ: r = 0.977, *p* < 0.001; XZ: r = 0.958, *p* < 0.001).

**Figure 4 polymers-12-00678-f004:**
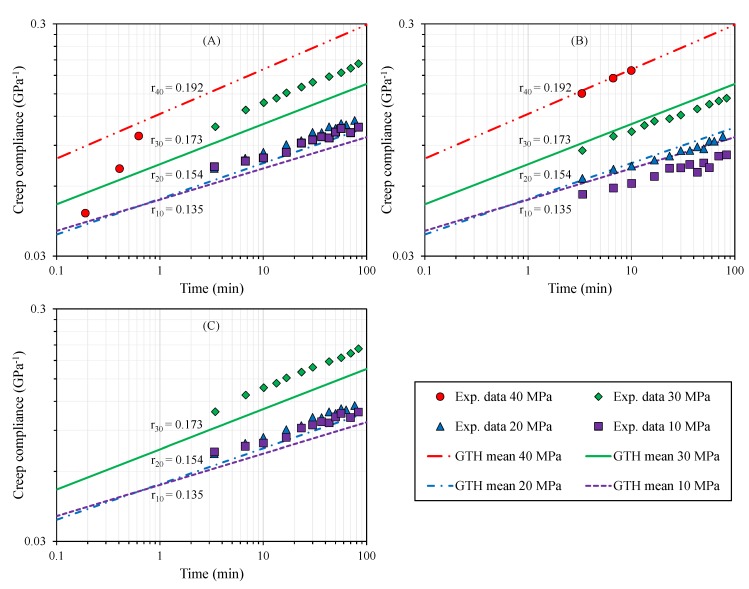
Creep compliance performance for different part build orientations at room temperature (RT): (**A**) XY, (**B**) YZ, and (**C**) XZ. XY and XZ were unable to reach 40 MPa. Averaged GTH coefficients (solid and dashed lines) showed the ability to model the creep compliance’s dependence on stress.

**Figure 5 polymers-12-00678-f005:**
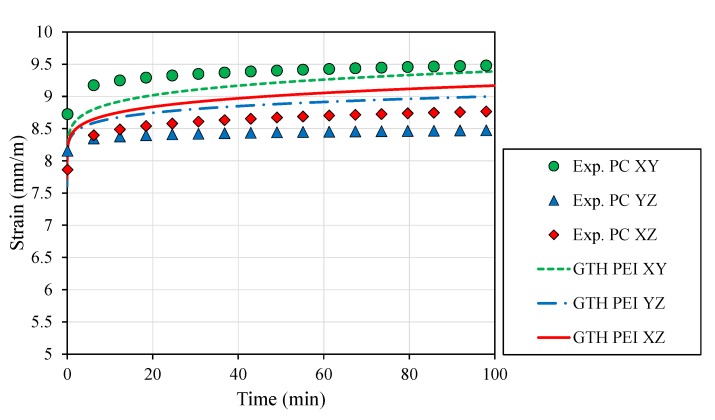
Creep curves for different part build orientations at RT for Polycarbonate (PC, markers) [27] and Polyetherimide (PEI, dashed lines) under 16 MPa. The GTH model was used with orientation-dependent coefficients to predict PEI behavior.

**Figure 6 polymers-12-00678-f006:**
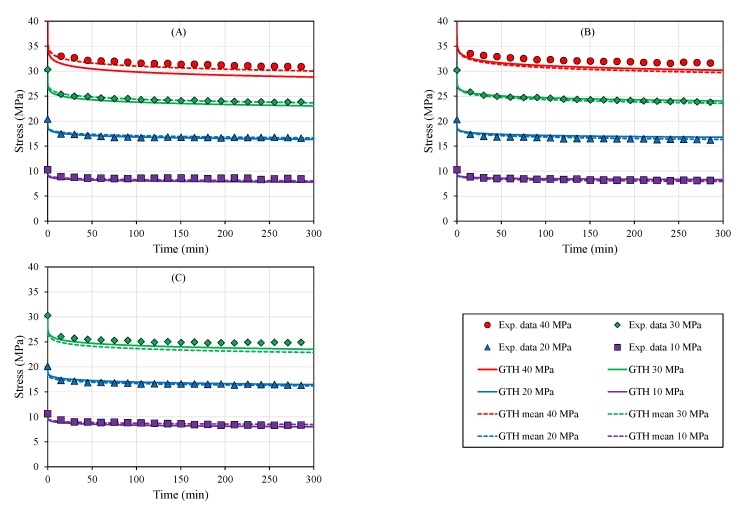
Stress relaxation curves for different part build orientations at RT: (**A**) XY, (**B**) YZ, and (**C**) XZ. The XZ orientation’s specimens broke during the initial loading to 40 MPa. The GTH coefficients fitted to each orientation (solid lines) were able to predict stress relaxation behavior in all orientations (XY: r = 0.998, *p* < 0.001; YZ: r = 0.999, *p* < 0.001; XZ: r = 0.998, *p* < 0.001), although the averaged GTH coefficients (dashed lines) provided a better predictive performance (XY: r = 0.999, *p* < 0.001; YZ: r = 0.999, *p* < 0.001; XZ: r = 0.998, *p* < 0.001).

**Figure 7 polymers-12-00678-f007:**
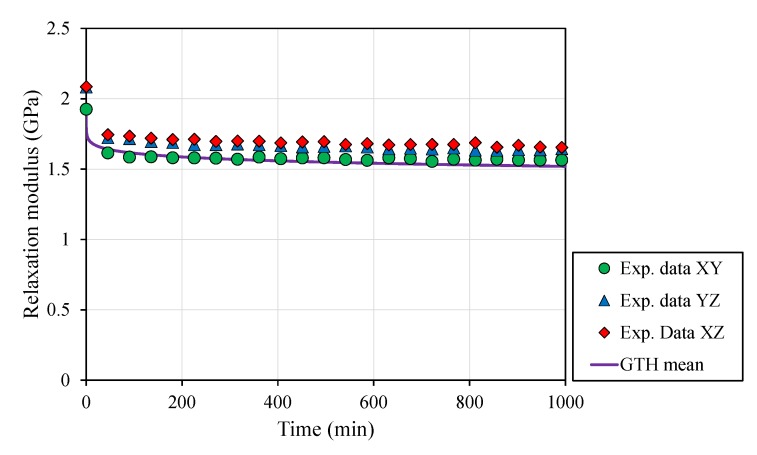
Stress relaxation curves for different part build orientations at RT: XY, YZ and XZ, along with the averaged GTH coefficients, proved the model’s ability to predict long time tests.

**Table 1 polymers-12-00678-t001:** Overview of previous creep and stress relaxation work in Fused-Deposition Modelling (FDM) technology by varying process parameters.

Reference	Processing Parameters	Materials
[23,24]	Slice thickness, air gap, raster angle, bead width, number of contours	PC-ABS
[25]	Part build orientation	ABS
[26]	Part build orientation	ABS plus
[27]	Part build orientation, air gap, number of contours	PC

ABS: Acrylonitrile butadiene styrene; PC: Polycarbonate.

**Table 2 polymers-12-00678-t002:** Fixed process parameters along with their nominal values.

Parameter	Value	Unit
Layer thickness	0.254	mm
Raster width	0.508	mm
Air gap	0.000	mm
Raster angle	45	degree
Number of contours	1	-
Contour to contour air gap	0.000	mm
Contour to raster air gap	0.000	mm
Contour width	0.508	mm
Part interior style	Solid	-
Visible surface style	Normal	-
Oven temperature	195	${}^{\circ }$C
Bed temperature	195	${}^{\circ }$C
Nozzle temperature for model material	390	${}^{\circ }$C
Nozzle temperature for support material	280	${}^{\circ }$C
Print speed	28	mm/s

**Table 3 polymers-12-00678-t003:** Sample design matrix for the creep tests.

Part Build Orientation	$\sigma_{1}$ (MPa)	$\sigma_{2}$ (MPa)	$\sigma_{3}$ (MPa)	$\sigma_{4}$ (MPa)	Loading Time /Recovery Time (min)
XY	10	20	30	40	100/400
YZ	10	20	30	40	100/400
XZ	10	20	30	40	100/400

**Table 4 polymers-12-00678-t004:** Sample design matrix for the stress relaxation tests. Loading time, in minutes, appears in parenthesis. Loading time in $\sigma_{2}$ was used to prove the validity of Equation (Equation 1) for longer time periods.

Part Build Orientation	$\sigma_{1}$ (MPa)	$\sigma_{2}$ (MPa)	$\sigma_{3}$ (MPa)	$\sigma_{4}$ (MPa)
XY	10 (300)	20 (1000)	30 (300)	40 (300)
YZ	10 (300)	20 (1000)	30 (300)	40 (300)
XZ	10 (300)	20 (1000)	30 (300)	40 (300)

**Table 5 polymers-12-00678-t005:** Elastic modulus, yield strength, ultimate strength, mass, and infill for each configuration.

Part Build Orientation	Elastic Modulus (GPa)	Yield Strength (MPa)	Ultimate Stress (MPa)	Mass (gr)	Infill (%)
XY	2.07 ± 7.9%	32 ± 5.3%	48 ± 9.0%	21.8 ± 0.12%	82.3 ± 0.12%
YZ	2.15 ± 7.7%	47 ± 2.6%	69 ± 12%	21.8 ± 0.13%	84.8 ± 0.13%
XZ	2.27 ± 9.8%	33 ± 9.7%	42 ± 9.0.%	22.0 ± 0.17%	94.3 ± 0.17%

YZ and XZ values are provided by Stratasys [35], except for mass.

**Table 6 polymers-12-00678-t006:** Strain recovery values for each orientation and each stress just before and after removing the stress and 400 min later. Strain is given in mm/mm.

Part Build Orientation	Level of Stress (MPa)	Creep Strain	Strain 400 min Later
	10	6.09$\;\times \;10^{-4}$	1.06$\;\times \;10^{-4}$
XY	20	2.14$\;\times \;10^{-3}$	2.72$\;\times \;10^{-4}$
	30	7.71$\;\times \;10^{-3}$	2.14$\;\times \;10^{-3}$
	10	7.69$\;\times \;10^{-4}$	1.03$\;\times \;10^{-4}$
	20	2.30$\;\times \;10^{-3}$	2.68$\;\times \;10^{-4}$
YZ	30	6.35$\;\times \;10^{-3}$	1.84$\;\times \;10^{-3}$
	40	1.59$\;\times \;10^{-2}$	5.37$\;\times \;10^{-3}$
	10	1.57$\;\times \;10^{-3}$	7.55$\;\times \;10^{-4}$
XZ	20	3.39$\;\times \;10^{-3}$	1.37$\;\times \;10^{-3}$
	30	9.98$\;\times \;10^{-3}$	3.21$\;\times \;10^{-3}$

**Table 7 polymers-12-00678-t007:** Adjusted parameters ($C_{x}$) for the generalized time hardening equation (see Equation (Equation 1)) along with the coefficient of determination ($R^{2}$). Coefficients must be multiplied by the exponential terms.

Part Build Orientation	$C_{1}$	$C_{2}$	$C_{3}$	$C_{4}$	$C_{5}$	$R^{2}$ for $\sigma_{1}$	$R^{2}$ for $\sigma_{2}$	$R^{2}$ for $\sigma_{3}$	$R^{2}$ for $\sigma_{4}$
XY	5.59$\;\times \;10^{-5}$	−2.93$\;\times \;10^{-6}$	8.23$\;\times \;10^{-8}$	1.07$\;\times \;10^{-1}$	2.55$\;\times \;10^{-3}$	0.928	0.980	0.991	-
YZ	4.40$\;\times \;10^{-5}$	−1.76$\;\times \;10^{-6}$	5.13$\;\times \;10^{-8}$	8.55$\;\times \;10^{-2}$	2.55$\;\times \;10^{-3}$	0.818	0.959	0.982	0.994
XZ	6.16$\;\times \;10^{-5}$	−4.16$\;\times \;10^{-6}$	1.07$\;\times \;10^{-7}$	1.33$\;\times \;10^{-1}$	1.70$\;\times \;10^{-3}$	0.903	0.966	0.991	-
Average	4.30$\;\times \;10^{-5}$	−1.80$\;\times \;10^{-6}$	5.31$\;\times \;10^{-8}$	1.15$\;\times \;10^{-1}$	1.92$\;\times \;10^{-3}$	-	-	-	-

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
