# Peer review of "Time-Dependent Mechanical Properties in Polyetherimide 3D-Printed Parts Are Dictated by Isotropic Performance Being Accurately Predicted by the Generalized Time Hardening Model"

_polymers, 2020, doi:10.3390/polym12030678_

Round 1
Reviewer 1 Report
Review of Time-dependent mechanical properties in Polyetherimide 3D-
printed parts are dictated by isotropic performance being accurately
predicted by the Generalized Time Hardening model
The manuscript is well-written and all steps are clearly indicated.
Unfortunately, there are fundamental issues in this work, so I would
suggest the editor to give authors another chance, if they are willing
to amend the work appropriately. In order to make the problem more
obvious, here are a few remarks:
1. FDM related materials and a long Introduction has been delivered. It
has been nicely and clearly written, with an emphasis on the structural
related anisotropy of (assumed to be) isotropic material. I find it
critical not to talk about the material property at the chain length
scale resulting in amorphous or semi crystalline material. Especially
in FDM, both are used, herein the chosen amorphous materials have to be
explained in such a way that possible readers realize that the layer
orientation is (presumably) the only approach to explain the
anisotropy. After this important point, later in the manuscript, all of
a sudden all direction related properties are used together as a mean
value (even reported that this works better). This fact is clearly
degrading the quality of the paper, especially after this long
Introduction with important emphasis on this relation.
2. After a long Introduction and a relatively detailed explanation
about material modeling, it is not easy to swallow Equation (1). Since
authors did cite Chaboche's work, I invite them to read it once more in
order to realize that strain and stress depend on each other (and their
rates) but not explicitly on time. In the case of a specific (herein,
relaxation) test, it may be the case that the solution looks like in
Equation (1), but this equation is a solution not a material equation.
The reasoning is very simple and can be found in all textbooks about
material modeling. If strain or stress depends explicitly on time, time
can be translated freely and changes the results. Equation (1) is the
reason that this manuscript has to be completely revised. None of the
results could be accepted.
3. As also indicated correctly, the process parameters are of
importance such that the Table 2 is fine. Please indicate also very
important features like the device and slicer used for manufacturing.
How many specimen were put on one bed? Any problems from one specimen
to the next one, is the excess material cut (I assume it is a
continuous production)? What is the ground level preparation, is it on
glass or aluminum, any special treatment used for?
4. Table 4 is not justified why the loading time for sigma_2 is
differently chosen than sigma_1, 3, and 4.
5. Figure 2 needs to be mirrored, stress-strain diagrams are given as
strain on x-axis. Also the explanation does not make sense, it is well-
known fact that manufacturer gives the data of the material measured by
standards for molding the material (there are no standards for 3-D
printing , yet). Mold parameters are of course slightly better (higher
strength) than the porous structure established by the FDM printer.
Authors writing a paper about FDM are expected to read the literature
before wondering why this case occurs in their study.
6. Table 6 is necessary, but it is misleading since strain "just before
and after removing the stress" is scientifically inadequate
information. Strain achieved at the removal needs to be given, not
right before and after, since the reader has no chance to find out what
authors really mean based on their time intuition indicated by "just
before."
Author Response
RESPONSES TO REVIEWER#1’s COMMENTS:
COMMENT 1: FDM related materials and a long Introduction has been delivered. It has been nicely and clearly written, with an emphasis on the structural related anisotropy of (assumed to be) isotropic material. I find it critical not to talk about the material property at the chain length scale resulting in amorphous or semi crystalline material. Especially in FDM, both are used, herein the chosen amorphous materials have to be explained in such a way that possible readers realize that the layer orientation is (presumably) the only approach to explain the anisotropy. After this important point, later in the manuscript, all of a sudden all direction related properties are used together as a mean value (even reported that this works better). This fact is clearly degrading the quality of the paper, especially after this long Introduction with important emphasis on this relation.
RESPONSE 1: Thank you for your comments. Reviewer is correct. Anisotropy in FDM production comes from the layer by layer deposition. We changed the sentences in this paper to clarify that we can provide a mean value model for strain accumulation and stress relaxation with an acceptable regression coefficient (lines 80-81; 175-177). This is important as real parts are manufactured with areas subjected to loads in many layer orientations.
“Significant anisotropic behaviour was found in FDM ABS specimens due to layer orientation during fabrication.”
“Therefore, the study of the parameters with the greatest influence on mechanical behaviour is necessary to understand the inner anisotropy of FDM parts arising from the layer orientation during fabrication.”
COMMENT 2: After a long Introduction and a relatively detailed explanation about material modeling, it is not easy to swallow Equation (1). Since authors did cite Chaboche's work, I invite them to read it once more in order to realize that strain and stress depend on each other (and their rates) but not explicitly on time. In the case of a specific (herein, relaxation) test, it may be the case that the solution looks like in Equation (1), but this equation is a solution not a material equation. The reasoning is very simple and can be found in all textbooks about material modeling. If strain or stress depends explicitly on time, time can be translated freely and changes the results. Equation (1) is the reason that this manuscript has to be completely revised. None of the results could be accepted.
RESPONSE 2: Thank you for pointing out this very relevant point. Prof. Garcia-Granada PhD thesis (University of Bristol, 2000) [1] describes in page 52 the difference in modelling time hardening versus strain hardening. Time hardening is only a mathematical approach that avoids numerical resolution of equations. Obviously material properties depend on previous strains and previous strain rates and from there you can obtain what happens to material after a certain time. This has been rephrased in the current version of the article (lines 138-143; 150).
“Furthermore, Garcia-Granada [33] discussed the possibilities of modelling creep. A material model based on previous strain and strain rates was discussed as strain hardening model and compared to a simplified model where stress and strain were function of time, named time hardening. This mathematical approach becomes indispensable for FEM simulations when parameters are obtained according to the true strain and strain rates.”
“The GTH model used in this approach is expressed as…”
“Because strain is constant over the time, it can be expressed as constant strain rate…”
COMMENT 3: As also indicated correctly, the process parameters are of importance such that the Table 2 is fine. Please indicate also very important features like the device and slicer used for manufacturing. How many specimens were put on one bed? Any problems from one specimen to the next one, is the excess material cut (I assume it is a
continuous production)? What is the ground level preparation, is it on glass or aluminium, any special treatment used for?
RESPONSE 3: Thank you for your comment. As stated in 3.1, specimens were manufactured in a Stratasys Fortus 400 mc machine. Slicer software Insight (OEM original software) was used to slice parts. Specimens were manufactured in groups of up to 40 in the same foundation sheet. Fortus 400 mc machine permits a traceable manufacturing process. It works with a temperature-controlled chamber. Parts are manufactured over a polymeric disposable foundation sheet fixed on an aluminium plane table of 355x254x254 mm dimensions. Original foundation sheets from OEM were used. There is no need of special treatment used to guarantee adherence between parts and foundation sheet. All parts in the same building group are manufactured with rigorous temperature and geometry control, and it is not expected to find substantial differences among specimens in the same manufactured group (please, see the added percentual values of specimens’ mass in Table 5, which values lower than 1%). A breakable support structure is used, when needed, to guarantee hanging surfaces geometry. Software automatically define supporting structures with low density deposition and as less contact as possible with manufactured surface. A material pack in a Fortus 400 mc machine has a volume of 1500 cubic centimetres. A group of 40 specimens needs a volume of less than 400 cubic centimetres. There was no difference in the material among specimens of the same group. This has been summarized in the main text (lines 163-166).
“Slicer software Insight was used to slice specimens, which were manufactured in groups of up to 40 in the same material and foundation sheet within a temperature-controlled chamber. Parts were manufactured over a polymeric disposable tray fixed on an aluminium plane table without special treatments.”
COMMENT 4: Table 4 is not justified why the loading time for sigma_2 is differently chosen than sigma_1, 3, and 4.
RESPONSE 4: Thank you for your comment. We tried to make it clearer in the current version of the document (lines 206-207), and in the caption of Table 4.
“This longer test enabled us to prove the validity of the equations presented above for longer periods.”
COMMENT 5: Figure 2 needs to be mirrored, stress-strain diagrams are given as strain on x-axis. Also the explanation does not make sense, it is well- known fact that manufacturer gives the data of the material measured by standards for molding the material (there are no standards for 3-D printing, yet). Mold parameters are of course slightly better (higher strength) than the porous structure established by the FDM printer. Authors writing a paper about FDM are expected to read the literature before wondering why this case occurs in their study.
RESPONSE 5: We swapped axis in Figure 2 to obtain a typical stress / strain relation. Regarding your latter point, any publication cited in the manuscript has been used ASTM standards to characterize FDM specimens. Furthermore, material’s manufacturer also manufactures the FDM machine, providing tensile data for printed specimens given specific part build orientations. Not injection moulding data is provided by the manufacturer.
COMMENT 6: Table 6 is necessary, but it is misleading since strain "just before and after removing the stress" is scientifically inadequate information. Strain achieved at the removal needs to be given, not right before and after, since the reader has no chance to find out what authors really mean based on their time intuition indicated by "just before."
RESPONSE 6: The reviewer was obviously right. Since the terminology is misleading, we have applied the following changes:
- Strain before removal column was deleted since it was the total elastic and creep strain. Since elastic strain is provided in figure 2 and creep strain in the next column, this one was redundant.
- Creep strain: no changes.
- Strain after removal: this column was deleted since no reference was made in the manuscript.
- Strain 400 min later: no changes.
References
[1] Garcia-Granada A-A. The effect of creep and mechanical load on cold expanded fastener holes. University of Bristol, 2000.
Reviewer 2 Report
Generally I consider the work as very good, however some minor issues should be adressed by Authors:
In the introduction section Authors could mention the relatively fresh work from other MDPI journal - Materials, related to the dynamic mechanical properties of ABS printed samples:
Static and Dynamic Mechanical Properties of 3D Printed ABS as a Function of Raster Angle
Materials 2020, 13(2), 297
Authors are writing that parameters presented in Table 2 "have been widely accepted as optimal parameters" regarding "dimensional accuracy, surface finish, and mechanical behaviour". However, data presented in the following research work state different way:
FDM process parameters influence over the mechanical properties of polymer specimens: A review
Polymer Testing, Volume 69, August 2018, Pages 157-166
A lot of cited works state that the best tensile performance was noted for raster angle of 0 degree. Authors should consider cited data and discuss that matter.
Data from mechanical tests should be presented with standard deviations.
Author Response
COMMENT 1: In the introduction section Authors could mention the relatively fresh work from other MDPI journal - Materials, related to the dynamic mechanical properties of ABS printed samples: “Static and Dynamic Mechanical Properties of 3D Printed ABS as a Function of Raster Angle, Materials 2020, 13(2), 297”.
RESPONSE 1: We agree with this point. The reference was added in lines 82-84.
“Domingo-Espin et 52 al. [7], Croccolo et al. [8] and Galeja et al. [9] studied the tensile properties of PC and ABS parts when varying the layer orientation, raster angle, and number of contours. “
COMMENT 2: Authors are writing that parameters presented in Table 2 "have been widely accepted as optimal parameters" regarding "dimensional accuracy, surface finish, and mechanical behaviour". However, data presented in the following research work state different way: “FDM process parameters influence over the mechanical properties of polymer specimens: A review, Polymer Testing, Volume 69, August 2018, Pages 157-166”. A lot of cited works state that the best tensile performance was noted for raster angle of 0 degree. Authors should consider cited data and discuss that matter.
RESPONSE 2: We understand your point. In this revised version we included changes to emphasize that the raster angle configuration provided in the manuscript is optimal for all the given part build orientations (lines 188-190). Obviously, attending to previous publications, when you manufacture parts with null raster angle, the mechanical properties are improved in that specific part build orientation, but the mechanical performance in other orientations is clearly diminished.
“Sood et al. [35,36] and Ziemian [11] showed that the values stated in Table 2 provide good results in terms of dimensional accuracy, surface finish, and mechanical behaviour in any given part build orientation.”
COMMENT 3: Data from mechanical tests should be presented with standard deviations.
RESPONSE 3: The reviewer is obviously right. The percentual deviation have been added to Table 5.
Round 2
Reviewer 1 Report
Further language checks are recommended.
Author Response
We already went through MDPI English editing services. See attached file.
